# Red Meat Derived Glycan, N-acetylneuraminic Acid (Neu5Ac) Is a Major Sialic Acid in Different Skeletal Muscles and Organs of Nine Animal Species—A Guideline for Human Consumers

**DOI:** 10.3390/foods12020337

**Published:** 2023-01-10

**Authors:** Marefa Jahan, Peter C. Thomson, Peter C. Wynn, Bing Wang

**Affiliations:** 1School of Agricultural, Environmental and Veterinary Sciences, Charles Sturt University, Wagga Wagga, NSW 2678, Australia; 2School of Life and Environmental Sciences, University of Sydney, Camden, NSW 2750, Australia; 3Gulbali Institute for Agriculture, Water and Environment, Charles Sturt University, Wagga Wagga, NSW 2678, Australia

**Keywords:** sialic acid, N-acetylneuraminic acid (Neu5Ac), ketodeoxynonulosonic acid (KDN), muscle and organ tissue, animal species

## Abstract

Sialic acids (Sias) are acidic monosaccharides and red meat is a notable dietary source of Sia for humans. Among the Sias, N-acetylneuraminic acid (Neu5Ac) and 2-keto-3-deoxy-D-glycero-D-galacto-2-nonulosonic acid (KDN) play multiple roles in immunity and brain cognition. On the other hand, N-glycolylneuraminic acid (Neu5Gc) is a non-human Sia capable of potentiating cancer and inflammation in the human body. However, their expression within the animal kingdom remains unknown. We determined Neu5Ac and KDN in skeletal muscle and organs across a range (*n* = 9) of species using UHPLC and found that (1) caprine skeletal muscle expressed the highest Neu5Ac (661.82 ± 187.96 µg/g protein) following by sheep, pig, dog, deer, cat, horse, kangaroo and cattle; (2) Among organs, kidney contained the most Neu5Ac (1992–3050 µg/g protein) across species; (3) ~75–98% of total Neu5Ac was conjugated, except for in dog and cat muscle (54–58%); (4) <1% of total Sia was KDN, in which ~60–100% was unconjugated, with the exception of sheep liver and goat muscle (~12–25%); (5) Neu5Ac was the major Sia in almost all tested organs. This study guides consumers to the safest red meat relating to Neu5Ac and Neu5Gc content, though the dog and cat meat are not conventional red meat globally.

## 1. Introduction

Sialic acids (Sias) are a diverse family of acidic, nine-carbon-backbone monosaccharides. They are ubiquitous in the animal kingdom and usually present as a conjugated sugar at the outermost (non-reducing) end of glycan chains of oligosaccharide structures of gangliosides, glycolipids and glycoproteins within the cell membrane and in secreted vertebrate glycoconjugates including those found in the human [1,2,3,4]. Sias are also part of glycosylated RNA where two kinds of sia-binding immunoglobulin-type lectins (Siglecs) can attach to the glycoRNAs of the outer cell membrane [5]. Sia rarely occurs as a free molecule [6] and comprises less than 3% of total Sia in mammalian brain [6,7]. Recently we reported the expression level of non-human Sia, N-glycolylneuraminic acid (Neu5Gc) in organs and skeletal muscles of nine animal species [8]. We established recommendations for consumers for the choice of animal meats to reduce the risk of cancer, as many studies reported that meat-derived Neu5Gc potentiates cancer progression in humans and is pro-inflammatory. In addition to Neu5Gc, the other predominant mammalian Sia is N-acetylneuraminic acid (Neu5Ac). Neu5Ac differs from Neu5Gc by a single oxygen atom, added to CMP-Neu5Ac in the cytosol, in a reaction catalysed by the enzyme cytidine monophosphate N-acetylneuraminic acid hydroxylase (CMAH) [9]. Humans are the only mammalian species unable to synthesise Neu5Gc, due to a chromosomal frameshift deletion in the CMAH gene: no other pathway has so far been identified. Therefore, humans cannot synthesise the Neu5Gc endogenously and it is not found in normal human tissues [9]. KDN (2-keto-3-deoxy-D-glycero-D-galacto-nononic acid), is a Sia prominently expressed in cold-blooded vertebrates. Free KDN was detected in cow milk products, porcine milk, human urine, and human ovarian and throat cancers [10]. KDN was found in human foetal newborn red blood cells at a 2–4 times higher concentration than found in the mother or healthy non- pregnant women [10]. This finding implies that, during embryogenesis KDN may be developmentally related to blood cell formation [10]. However, KDN-glycans are yet to be reported in normal human tissues [10].

Humans obtain their Sia Neu5Ac and Sia Neu5Gc from dietary sources. They also generate antibody responses against ingested Neu5Gc; thus, humans have circulating anti-Neu5Gc antibodies which are known as “xeno-auto-antigens”. As a result, Neu5Gc can facilitate the progression of cancer in humans [11]. The cancerous tissues are hypersialylated, and also contain more Neu5Gc than non-cancerous tissues [11]. However, dietary Neu5Ac can be incorporated into brain gangliosides, glycoproteins and polysialic acid (PolySia) which contributes to the overall net negative charge on the cell membrane, facilitating the binding of positively charged neurotransmitters [12]. Neu5Ac is the major form of Sia in all mammalian brains [13] and is an essential nutrient for brain development and cognition [14]. Sia is one of the most important molecules of life, with implications for uniquely human features both in health and disease [15].

Some meats are consumed more than others around the world, with red meat providing important micro- and macronutrients for human health. Animal organ meat is also consumed world-wide providing a rich source of iron, zinc and vitamins A, B2 and D [16]. Humans also derive Sia Neu5Ac from consumption of red meat. However, very limited information is available on the concentration of Neu5Ac and KDN and the relative proportions of Neu5Ac, Neu5Gc and KDN in red meat and organ meat from different animal species. Therefore, the objective of the present study was to determine the concentrations of free and conjugated forms of Neu5Ac and KDN in skeletal muscle and different organ tissues from nine animal species and to determine the proportion of Neu5Ac, Neu5Gc and KDN to serve as a guideline for dietary red meat choice to promote human health. We hypothesised that, like Neu5Gc, the concentrations of Neu5Ac and KDN are also tissue and species specific.

## 2. Materials and Methods

### 2.1. Sample Collection

All samples were collected from the Preclinical Veterinary Laboratory of Charles Sturt University, NSW, Australia. About 10–50 g of tissue for each sample from sheep (*Ovis aries*, *n* = 10), cattle (*Bos taurus*, *n* = 3), goat (*Capra aegagrus hircus*, *n* = 3), pig (*Sus scrofa*, *n* = 3), deer (*Odocoileus virginianus*, *n* = 3), horse (*Equus caballus*, *n* = 3), kangaroo (*Macropus giganteus*, *n* = 3), cat (*Felis catus*, *n* = 3) and dog (*Canis lupus familiaris*, *n* = 3) were collected. The experimental protocol was approved by The Animal Care and Ethics Committee of Charles Sturt University, Wagga Wagga, NSW, Australia (SL100018).

In case of sheep, muscle samples were obtained from 9 primal cuts, namely the leg chump, hind shank, short loin, eye of rack, tenderloin, fore shank, forequarter, neck muscle and peritoneal muscle. These tissues were used to determine if there were any differences in Neu5Ac and KDN concentrations from different muscles of the carcass that might help explain their possible functionality in metabolism. For the other animal species, muscle tissue was obtained from the thigh region. Heart, liver, lung, kidney, spleen and fat tissues were collected from each of the 9 species. Other edible organs, such as brain (*n* = 3), rumen, reticulum, omasum, abomasum, small intestine, large intestine and tongue were also collected from mature aged sheep. All these samples were stored at −20 °C pending analysis. Further details on the animals used in this study are provided in Appendix A.

### 2.2. Preparation of Muscle and Organ Tissues (Except Brain)

The muscle and organ tissues were processed as described previously [8,17] by using 100 mg of muscle or organ tissue. The tissue was homogenised with 1 mL of water using an Ultra Turax homogeniser (T25 basic, Janke & Kunkel IKA Labortechnik, Rawang Malaysia) at speed 4 (19,000 rpm) for 3 min. With 100 µL homogenised tissues, 1 mL of 0.1 M NaOH (Sigma Aldrich, St. Louis, MO, USA) was added, mixed and incubated at 37 °C for 30 min to remove O-acetyl groups. After incubation 1.22 mL of 0.1 N HCl (Sigma Aldrich, St. Louis, MO, USA) was added to neutralise the solution (was tested with pH paper strip to confirm neutralisation). To the neutralised solution, 0.27 mL glacial acetic acid was added to reach the required final 2 M acetic acid concentration. Half of this sample solution was heated at 80 °C for 3 h to release conjugated Sia before analysis using UHPLC [8]

### 2.3. Preparation of Brain Sample

The brain contains no actual muscle other than the middle layer of the arteries that carry blood to the brain. Nearly 60% of human brain tissue is fat, making it the fattiest organ in the body. Therefore, to extract/isolate most glycans from the tissues we used different preparation methods for the brain.

The preparation of brain sample for UHPLC analysis was undertaken using our published method [8]. Briefly, brain (frontal cortex, 50 mg) sample was homogenised (Ultra Turax homogeniser) with 150 µL distilled water, then 525 µL methanol (Sigma Aldrich, St. Louis, MO, USA) was added while continuously shaking. Then chloroform (270 µL; Sigma Aldrich, St. Louis, MO, USA) was added, mixed for 2 min and the mixture was centrifuged at 2000× *g* for 30 min at 4 °C with the supernatant being discarded. The bottom pellet was then homogenised in 100 µL water followed by the addition of 400 µL chloroform: methanol (1:2, *v*:*v* and centrifugation at 3000× *g* for 30 min at 4 °C. The supernatant was removed. The upper 2 layers were combined, with the pellet being used for protein-conjugated sialic acid analysis. The combined upper layers received 313 µL water providing a chloroform: methanol: water ratio of 1:2:1.4. This mixture was then centrifuged at 3000× *g* for 10 min at 4 °C. The upper layer was removed, while 150 µL methanol and then 100 µL of 0.01 M KCL (Sigma Aldrich, St. Louis, MO, USA) was added slowly to the bottom layer, agitated for 2 min and centrifuged at 3000× *g* for 15 min at 4 °C. The 2 upper layers were combined, and 10 µL 1-butanol was added as an anti-foaming agent. Each sample was then freeze-dried (CHRiST, John Morris Scientific Pty. Ltd., Osterode am Harz, Germany) and re-dissolved in 200 µL chloroform:methanol (1:1, *v*:*v*). This solution was then sonicated (POWER SONIC410, Hwashin Technology Co, Seoul, Republic of Korea) on ice for 15 s (5 s × 3 times) and retained for ganglioside Sia analysis. In order to estimate protein-conjugated Sia, 200 µL 0.05 M H2SO4 was added, sonicated for 15 s and then heated for 1 h at 80 °C. Maximum release of ganglioside—Conjugated Sia was achieved in 0.1 M trifluoroacetic acid (Sigma Aldrich, St. Louis, MO, USA) heated for 150 min at 80 °C [18]. Subsequent to hydrolysis, 200 µL water was added to the protein fraction. Both the protein and ganglioside fractions were filtered separately through a 0.22 μm Millipore membrane filter (Millex^®^ Merck Millipore, Darmstadt, Germany) prior to quantitation of sialic acid using UHPLC.

### 2.4. Preparation of Mixed Sia Standards of Neu5Ac and KDN

Standards of Neu5Ac and KDN were obtained from Sigma Aldrich, St. Louis, MO, USA. Ten mg of each dried standard (Neu5Ac—Cat no. A0812 and KDN—Cat no. 60714) was dissolved in 1 mL of deionised water to prepare 10 mg/mL stock solutions. Then, 2 µL 10 mg/mL Neu5Ac and 1 µL 10 mg/mL KDN was mixed in 0.997 µL deionised water. This solution was then serially diluted to prepare a set of standards.

### 2.5. DMB Method for Derivatisation of Neu5Ac and KDN Standards and Experimental Samples

For quantification of both conjugated and free Sia, DMB (1,2-diamino-4,5-methylenedioxybenzene, Sigma Aldrich, St. Louis, MO, USA, Cat no. 66807) reagent was used. The DMB reagent was prepared according to the method of Inoue [18] as follows: 14 mM DMB, 18 mM sodium hydrosulphite (Sigma Aldrich, St. Louis, MO, USA, Cat no. 71699), 1 M 2-mercaptoethanol (Sigma Aldrich, St. Louis, MO, USA, Cat no. M6250) and 40 mM trifluoroacetic acid (Sigma Aldrich, St. Louis, MO, USA, Cat no. 302031). A total of 50 µL DMB reagent was added to 50 µL of filtered sample or standard (10 µL standards and 40 µL diluents) solution. To protect from light, these preparations were done in a brown HPLC vial using an insert (Agilent Technologies, Santa Clara, CA, USA). All samples were mixed and subsequently incubated at 4 °C for 48 h [17]. Following the derivatisation by DMB, the samples were assessed for the concentration of the Sia by UHPLC (see Section 2.7).

### 2.6. Standard Curve for Neu5Ac and KDN

Standard curves for Neu5Ac were constructed in the concentration range of 0.004–1 ng/µL while, for KDN, standard curves were constructed in the concentration range of 0.002–0.5 ng/µL (Appendix A)

### 2.7. UHPLC Analysis

The measurement of Neu5Ac and KDN was undertaken with our validated published method [19]. In brief, after being derivatised with DMB, all samples were analysed with an Agilent 1290 Infinity system (Agilent Technologies, Santa Clara, CA, USA) using a 1260 Infinity Fluorescence Detector (Agilent Technologies, Santa Clara, CA, USA). The column (Zorbax SB-Aq Rapid Resolution column-3.5 μm, 4.6 × 50 mm, Agilent Technologies, Santa Clara, CA, USA) was held at 30 °C and the auto sampler was held at a constant 4 °C. Elution of Sia from the derivatised samples was performed in the column using water: methanol: acetic acid (75:25:0.05 *v*/*v*/*v*) with flow rate of 1.5 mL/min for 3.5 min. The columns were washed with a mixture of methanol: water: acetic acid (80:20:0.05 *v*/*v*/*v*) for 1.4 min before equilibration for 4 min (total run time of 9 min). Fluorescence was detected through emissions at 448 nm s and 373 nm excitation. All samples were analysed in duplicate with injection volume of 10 μL and concentrations were expressed as µg/g for each sample. The coefficient of variation (CV) for UHPLC analysis within the same day and within different days was 2–5%.

### 2.8. Determination of the Level of Protein in Different Tissue Organs

Total protein concentration was assessed in each homogenate with the Pierce BCA protein assay kit (23,225, Thermo Scientific, Waltham, MA, USA). The concentrations of free and conjugated Sias were normalised to the protein content and expressed as μg Sia/g protein for all samples with the exception of fat tissue. All samples were assayed in duplicate.

### 2.9. Quantitative Determination of Total Lipid in Fat Tissue

Fat content was assessed by extraction of samples in hexane using a Soxtec 2050 apparatus (FOSS Analytical, Hoganas, Sweden) according to AOAC 2003.06. In brief, 0.50–1.00 g of the freeze-dried ground sample was weighed and added to a cellulose thimble (Whatman 33 mm × 80 mm double thickness, China) and then placed in a beaker. The sample was then oven-dried (Contherm Thermotec 2000, Wellington, New Zealand) for a further 2 h at 105 °C to remove any additional moisture and the thimbles were then placed in the Soxtec extraction apparatus. The corresponding beakers were then rinsed with 70–90 mL of hexane into a pre-dried and pre-weighed aluminium cup containing small glass beads. The extraction proceeded at 155 °C with boiling for 20 min, rinsing for 40 min and recovery for 10 min. Upon completion, the aluminium cups were placed in a drying oven (Contherm Thermotec 2000, Wellington, New Zealand) at 105 °C for 20–30 min, and cooled in a desiccator (Custom made, containing Silica Gel, Sigma Aldridge P/N 10087), prior to weighing and calculating the fat content (%). The final concentration of free and conjugated Sias in fat tissue was normalised to the lipid concentration and expressed as μg Sia/g lipid.

### 2.10. Statistical Analysis

Results for total Sia, Neu5Ac and KDN in different organ tissues from nine species were analysed using linear mixed models. Species and sample name (muscle or organ) were used as the fixed effects, as well as their interaction to assess whether sample effects differed between species. Animal number was set as a random effect. Residual plots were used to ensure that the model assumptions were met for all models. With the exception of sheep, comparisons between organs within the same species were made by least significant differences (LSDs) when the overall F-test was significant. Comparisons between species for the same organ were also compared using LSDs. The analysis was conducted using GenStat statistical software (17th Edition).

## 3. Results

### 3.1. Comparison of Concentration of Total, Conjugated and Free Neu5Ac in the Organs and Skeletal Muscle of Nine Different Animal Species

Both conjugated and free Neu5Ac were detected in all organs and skeletal musculature assessed from the nine species used in this study. The mean total Neu5Ac concentration in skeletal muscle was about ~343 µg/g protein in nine animal species (Figure 1A). The rank order of total and conjugated Neu5Ac in skeletal muscle from high to low was goat, sheep, pig, dog, deer, cat, horse, kangaroo and cattle (Figure 1A). Kidney contained the highest total Neu5Ac (1.99–3.05 mg/g protein) in most animal species, except horse, kangaroo and cat, in which lung was the highest (Figure 1B). The lung contained the second highest total Neu5Ac in most animal species (Figure 1C). The concentration of total Neu5Ac in kidney or lungs was 4–18 times higher than the muscle in nine animal species. After kidney and lung, the rank order of total Neu5Ac concentration from high to low was liver, spleen and heart in all animals, except deer and kangaroo where spleen contained the higher concentration of total Neu5Ac compared to liver (Figure 1D–F). Fat tissue showed the lowest Neu5Ac content of all tested organ tissues of the nine animal species (Figure 1G). Significant differences in the same organ of nine tested species was observed only for heart (*p* < 0.001, Figure 1D) and fat tissue (*p* = 0.024, Figure 1G). However, the total Neu5Ac concentration across different organ tissues of the same animal species was significantly different (*p* < 0.001), except for the cat (*p* = 0.066) (Appendix A).

The concentration of conjugated Neu5Ac (118.86–4007.80 µg/g protein) was 3–39 times higher than free Neu5Ac (4.04–448.09 µg/g protein) including that in fat (14.90–50.55 µg/g lipid for conjugated and 0.05–7.23 µg/g lipid for free), based on organ tissue and animal species. In the horse, the conjugated Neu5Ac in lung (4007.80 µg/g protein), spleen (417.38 µg/g protein) and fat tissue (14.90 µg/g lipid) were 55, 103 and 270 times higher than free Neu5Ac, respectively. Overall, conjugated Neu5Ac was 70–97% of total Neu5Ac. Surprisingly, free Neu5Ac (177.32 µg/g protein) in cat muscle was higher than conjugated Neu5Ac (118.86 µg/g protein). In dog muscle, however, the concentration difference of free and conjugated Neu5Ac was quite small (conjugated form 202.75 µg/g protein and free form 155.08 µg/g protein) (Figure 2). The concentration of total and conjugated Neu5Ac was significantly different across the nine animal species (*p* = 0.023), but not with the free Neu5Ac (*p* = 0.784), for the species–sample interaction.

### 3.2. Comparison of the Concentration of Total, Conjugated and Free KDN in Organs and Skeletal Muscle of Nine Different Animal Species

Significant species differences for the total KDN concentration were observed for most of the organ tissues including the lung (*p* = 0.003, Figure 3C), liver (*p* = 0.051, Figure 3D), heart (*p* = 0.003, Figure 3E) and spleen (*p* < 0.001, Figure 3F), but not in muscle (*p* = 0.263, Figure 3A), kidney (*p* = 0.268, Figure 3B) or fat tissue (*p* = 0.136, Figure 3G). The highest content of total KDN was detected in the organ tissues of deer and the lowest concentration was found in those from the horse (Appendix A).

Unlike Neu5Ac, the conjugated form of KDN was negligible or absent in most of the organ tissues, while the free KDN (70–100% of total KDN) was the predominant form in all tested tissues (Figure 4). The only exception was the muscle of goat and liver and fat tissue of sheep, in which the conjugated KDN was 3–7 times higher than free KDN (Figure 4). Another interesting finding was that, although skeletal muscle of most animal species studied contained negligible concentrations of free KDN (0–0.01 µg/g protein), the skeletal muscle in horse had the highest concentration of free KDN (7.71 µg/g protein) followed by the pig (4.80 µg/g protein) (Figure 3A). We demonstrated that there was a significant difference in total KDN, conjugated KDN and free KDN concentration for seven different organ tissues within nine different species (*p* < 0.001, Appendix A). However, the highest KDN content was detected in the spleen of cattle, sheep and kangaroo; liver of goat, deer and dog; kidney of cat; heart of pig and muscle of horse (Appendix A).

### 3.3. Mean Proportion of Total Neu5Ac, Neu5Gc and KDN in the Organs, Skeletal Muscle and Fat Tissue of Different Animal Species

The mean proportion of Neu5Ac, Neu5Gc (the data for Neu5Gc were derived from Jahan, Thomson, Wynn and Wang [8], and the concentrations of Neu5Ac and KDN was determined in the same samples and at the same time as Neu5Gc concentration [8]) and KDN in percentage in the organ tissues of nine animal species is shown in Figure 5. The major Sia in the organ tissues, skeletal muscle and fat tissue of different animal species was Neu5Ac. Overall, the mean proportion of Neu5Ac varied from 60–90%, Neu5Gc varied from 10–40% and KDN was ˂ 1% depending on organ tissue and animal species (Figure 5). However, some animals such as the kangaroo, dog and deer contained an even higher percentage of Neu5Ac (85–99%) compared to Neu5Gc (15–˂1%). Among all organ tissues, muscle had the lowest proportion of Neu5Gc (10–20%), except for cattle where the proportion was 40%. In fat, the proportion of Neu5Gc was 3–30%, except for fat from cattle (56%). Kangaroo and dog muscle contained ~100% Neu5Ac, and no Neu5Gc. On the other hand, the spleen of cattle, pig, horse and cat contained a higher proportion of Neu5Gc at levels of 85% (cattle), 75% (pig and horse) and 60% (cat). Cat lung also contained a marginally higher proportion of Neu5Gc (51%) compared to Neu5Ac (49%).

In general, organ tissues of cattle had a relatively higher Neu5Gc content (52–85%) compared to those of other animals. Conversely, in kidney and muscle of cattle, the proportion of Neu5Ac (60%) was higher than that of Neu5Gc (40%). However, cattle lung, heart, liver and fat tissue contained lower proportions of Neu5Ac relative to Neu5Gc by 39%, 48%, 33% and 44%, respectively. The two female deer contained only Neu5Ac in all tested organs and were lacking any Neu5Gc except in the muscle [8].

### 3.4. Neu5Ac Content of Selected Skeletal Muscles and Sheep Visceral Organs

In sheep (*n* = 10), the concentrations of conjugated and free forms of Neu5Ac and KDN were analysed in muscles from nine different commercial primal cuts in the body and eight additional organ tissues (Table 1). Sheep were the species of choice because they were more readily available than other species as were their commercial primal cuts. There was no significant difference in conjugated and free Neu5Ac and KDN across the muscles of nine different primal cuts (*p* > 0.05). However, the difference was significant among eight additional organ tissues for both free and conjugated Neu5Ac and KDN (*p* ˂ 0.003–0.001, Table 1B). The highest concentration of conjugated Neu5Ac was detected in brain, while the highest free Neu5Ac was evident in the liver. The rank order of conjugated Neu5Ac concentrations from high to low was found in brain, lung, kidney, abomasum, small intestine, spleen, omasum, large intestine, liver, heart, reticulum, rumen, muscle and tongue. The conjugated form of Neu5Ac was higher than that of the free form in all tested organs and skeletal muscles. In contrast, the concentration of free KDN was higher than its conjugated form which was negligible in most organ tissues. The only exception was the liver, in which the concentration of conjugated KDN was higher than that of free KDN. The highest concentration of total KDN was evident in liver followed by large intestine, lung, small intestine, omasum, abomasum, spleen, heart, reticulum, kidney, tongue, rumen and muscle.

All the ganglioside-conjugated Sia in sheep brain consisted of Neu5Ac, with its concentration being higher than the 62% found in protein-conjugated Sia. However, no KDN was detected in any of these ganglioside or protein fractions.

## 4. Discussion

Red meat provides a balanced source of amino acids in protein, vitamin B12, iron, zinc and other trace minerals, and various other nutrients. Therefore, it is one of the most nutritious foods for human consumption. However, distribution and concentration of Neu5Ac and KDN in red meat of different animal species is not known. Recently we reported the non-human Sia glycan, Neu5Gc, was not identified in many organs and skeletal muscles of nine animal species [8]. This was an important finding with reference to their consumption by humans since red meat derived Neu5Gc promotes inflammation and progression of cancer [17]. In our present study we found that all tested organs and skeletal muscles contained Neu5Ac, but negligible/no KDN. Neu5Ac in the organs was expressed at a higher concentration than the skeletal muscles. Sheep brain contained a higher proportion of conjugated Neu5Ac than Neu5Gc, but not any conjugated KDN. This is consistent with the established role for Neu5Ac in brain cognitive function [13]. The majority of Neu5Ac in organ tissues across species was in the conjugated form (>70%). However, in the case of KDN, the free form was predominant (~70–100%). Sias are found to occur mostly as a conjugated sugar at the terminus of glycan chains (oligosaccharide structures of glycolipids and glycoproteins), usually covalently linked by an α-glycosidic bond. The conjugated Sia is the bioavailable form of Sia [17] with a broad spectrum of important functional roles in mucus viscosity, proteolytic protection of proteins, cell–cell recognition, reproduction, infection, immunity and cognitive development [20].

Sia is found rarely as a free molecule [6]. The biological significance of free Sia is still under investigation. Iijima et al. (2004) reported that under physiological conditions free Sia neutralised toxic hydrogen peroxide (H_2_O_2_) [21] through a radical reaction mechanism. This reaction requires the formation of a decarboxylated metabolite, 4-(acetylamino)-2,4-di-deoxy-D-glycero-D-galacto octonic acid (ADOA) from the free reducing end of the sugar [22]. The reaction can also occur between free Sia and related biologically important lipid hydroperoxides [23]. Lipid hydroperoxides are involved in oxidative stress through reactive oxygen species (ROS), which in turn give rise to inflammation including arteriosclerosis [23]. Studies in Wistar rats also showed that free Sia inhibited lipopolysaccharide (LPS)-induced renal failure, as well as LPS-induced liver dysfunction by neutralising ROS-mediated oxidative damage [24,25].

Although the biological functions of KDN are not fully documented, surprisingly high levels of free KDN are observed in many human cancers [26] and it has been suggested that free KDN can be a sensitive biomarker for early cancer diagnosis [27]. In the current study, we detected considerable concentrations of conjugated and free KDN in different organ tissues of the nine tested animal species, but the KDN expression levels were found to be both tissue and species specific. To explore the molecular basis of this finding further research is needed. It is also of critical importance to identify any negative impact of KDN consumption through red meat on human health. Recently healthy human sera have been shown to contain naturally occurring antibodies against the KDN glycans similar to Neu5Gc antibodies [28]. Nevertheless, the prevalence of anti-KDN antibodies was considerably lower than those against Neu5Gc glycans [28]. So, the role of any possible KDN incorporation on the human tissues also needs to be studied further.

In this study, we quantified the concentration of Neu5Ac and KDN in raw organ tissue only. Samraj, Pearce, Läubli, Crittenden, Bergfeld, Banda, Gregg, Bingman, Secrest and Diaz [17,26] found that cooked meat contained higher concentrations of Neu5Ac and Neu5Gc (~2-fold higher) compared to raw meat and based on different cooking methods (baked, boiled, fried). However, there was little change in the proportion of Neu5Ac to Neu5Gc between raw meat (55%) and cooked meat (52–53%, except in skeletal muscle where the proportion of Neu5Gc increased by 37% after cooking. Therefore, cooking method may affect differentially the concentration of various forms of Sia in the cooked product which may be related to differences in the lability of α2,3, α2,6 and α2-8-glycosidic linkages in the glycans. Additionally, there may be some variation in the degree of protein denaturation [26] depending on cooking temperature and moisture content. Thus, the ratio of Neu5Ac, Neu5Gc and KDN to protein in different organ tissues may vary with cooking method. Further study is required to explore the actual mechanism for the variation in expression levels of Neu5Ac and Neu5Gc with different cooking methods.

Our findings are consistent with the recent pig study reported by Ji, Wang, Chen, Yang, Zhang, Zhang, Troy and Wang [26] who assessed the concentration of conjugated and free Neu5Ac and KDN in organs and skeletal muscles of varying aged pigs by LC-MS/MS. The major form of Sia in pig organ tissue (spleen, kidney, lung, heart, liver and skeletal muscle) was Neu5Ac, with spleen being the exception containing ~75–80% Neu5Gc, which is consistent with our present study.

Red meat has been an indispensable part of the human diet, and global meat consumption has increased by 58% over the 20 years to 2018 [29]. Red meat derived Sia Neu5Ac in the human diet has many beneficial biological functions, in particular immuno-modulation and promotion of cognitive performance. However, not all red meat derived Sias are beneficial for humans. We documented the concentration and distribution of the non-human Sia Neu5Gc in the muscle and organ tissue of these nine animal species in our recent study [8]. The red meat derived Sia Neu5Gc is thought to act as a risk factor for human cancers and cardiovascular diseases. Therefore, the World Cancer Research Fund and the American Institute for Cancer Research recommend a limitation on consumption of red meat to about 350–500 g (about 12–18 oz) cooked weight per week [30]. Nevertheless, red meat is a valuable source of nutrients, containing the higher concentration of beneficial Sia Neu5Ac compared to Neu5Gc.

We have reported the proportion Neu5Ac, Neu5Gc and KDN in the organs and skeletal muscle across the selected range of animal species (Figure 5). Theoretically, red meat without Neu5Gc including female deer organs, and muscles from kangaroo and dog, should be the first choice for human consumption. All muscles from domestic livestock animals including cattle, sheep, pig and goat contained relatively higher proportions of Neu5Ac compared to their organ meats; therefore on this basis their musculature is highly recommended for human consumption. However, neither dog nor cat meat is a conventional red meat consumed globally though there are variations in some cultures.

The similar concentrations of Sia Neu5Ac across nine different commercial ovine muscle meat cuts reported here have shown that higher valued cuts do not provide any additional nutritional advantage for the consumer. We also found Neu5Ac was the major form of Sia across the spectrum of organ tissues and animal species with the exception of the spleen of cattle (~85% Neu5Gc), pig (~75% Neu5Gc), horse (~75% Neu5Gc) and cat (~60% Neu5Gc). Therefore, these organs must be avoided for human consumption.

Red meat is an excellent source of protein, iron, zinc, vitamin B12, niacin, and creatine, but is also the main source of Neu5Ac for human consumption. The concentration and proportion of Neu5Ac, Neu5Gc and KDN in red meat was both species and tissue specific. The results of this study will assist consumers in making a more informed decision on their choice of red meat based on concentrations of both health-giving Sia Neu5Ac and the pro-inflammatory Sia Neu5Gc.

## Figures and Tables

**Figure 1 foods-12-00337-f001:**
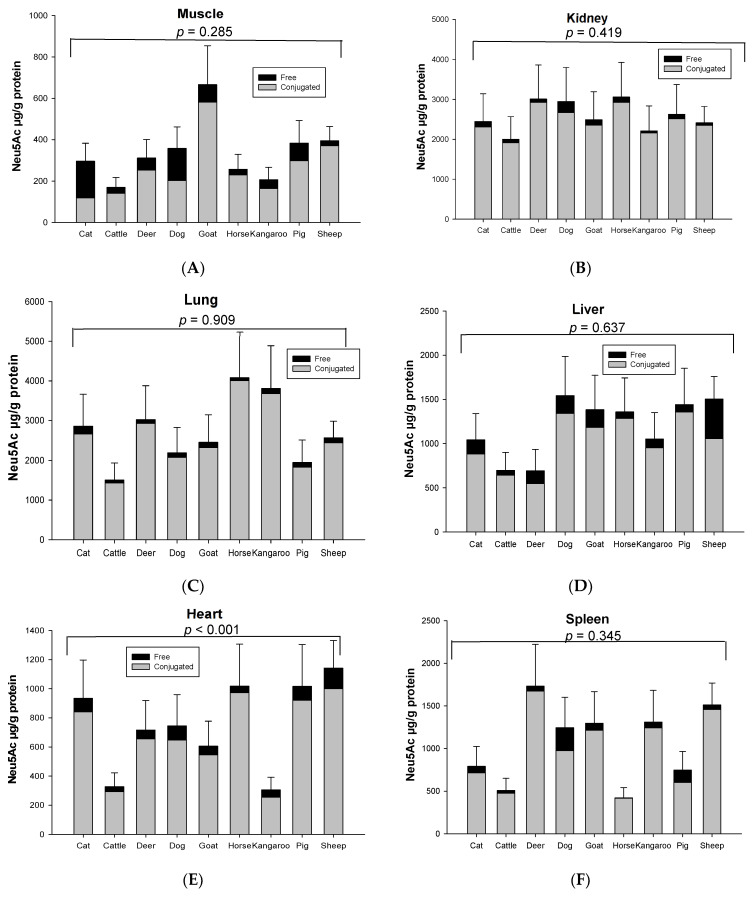
Concentration of total conjugated and free Neu5Ac in the same organ of nine tested animal species (*n* = 10 for sheep and *n* = 3 for other animal species)—(**A**) Muscle, (**B**) Kidney, (**C**) Lung, (**D**) Liver, (**E**) Heart, (**F**) Spleen and (**G**) Fat tissue. Values are mean ± SEM.

**Figure 2 foods-12-00337-f002:**
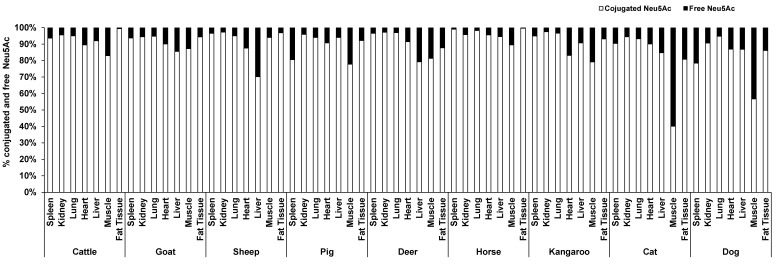
The average proportion of conjugated and free forms of Neu5Ac in the seven organ tissues of nine different animal species (*n* = 10 for sheep and *n* = 3 for other animal species).

**Figure 3 foods-12-00337-f003:**
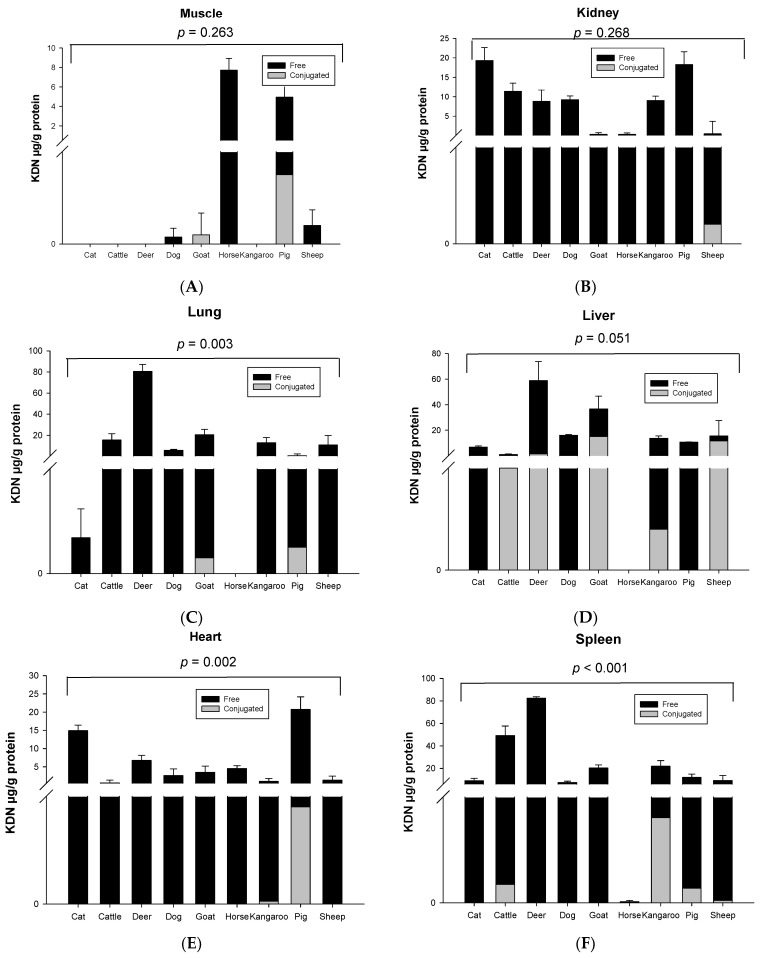
Concentration of total KDN in the same organ of nine tested animal species (*n* = 10 for sheep and *n* = 3 for other animal species)—(**A**) Muscle, (**B**) Kidney, (**C**) Lung, (**D**) Liver, (**E**) Heart, (**F**) Spleen and (**G**) Fat tissue. Values are mean ± SEM.

**Figure 4 foods-12-00337-f004:**
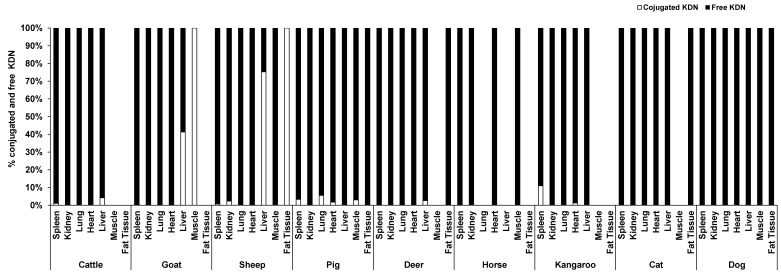
The average proportion of conjugated and free forms of KDN in the seven organ tissues of nine different animal species (*n* = 10 for sheep and *n* = 3 for other animal species).

**Figure 5 foods-12-00337-f005:**
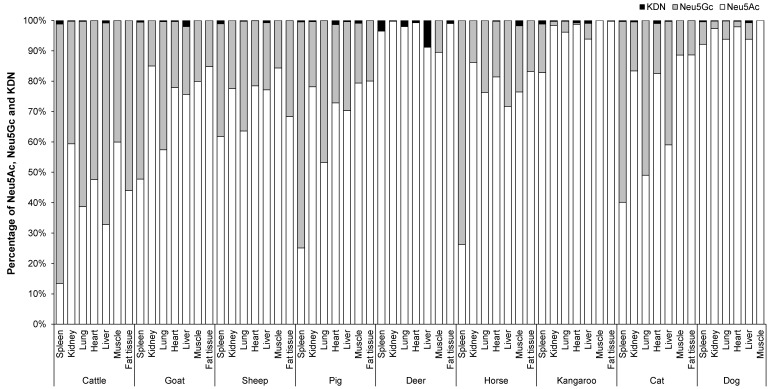
The proportion Neu5Ac, Neu5Gc and KDN in the organs and skeletal muscle of nine different animal species (*n* = 10 for sheep and *n* = 3 for other animal species).

**Table 1 foods-12-00337-t001:** Concentrations of Neu5Ac and KDN (µg/g protein) in muscles selected from different commercial primal cuts (**A**) and the additional eight organs of sheep ^1^ (**B**).

A				
Commercial Primal Cut	Neu5Ac		KDN	
	Conjugated Form	Free Form	Conjugated Form	Free Form
Hind Shank	275.89 ± 113.67	37.68 ± 16.88	0 ± 0	0.02 ± 0.02
Leg chump	370.93 ± 152.82	23.38 ± 10.48	0 ± 0	0.04 ± 0.04
Tenderloin	140.05 ± 57.70	63.31 ± 28.36	0 ± 0	0.02 ± 0.02
Short loin	346.89 ± 142.92	62.12 ± 27.83	0 ± 0	0.02 ± 0.02
Eye of Rack	237.70 ± 97.93	39.29 ±17.60	0 ± 0	0 ± 0
Forequarter	292.95 ± 120.70	44.70 ± 20.03	0 ± 0	0.02 ± 0.02
Fore shank	351.08 ± 144.64	61.07 ± 27.36	0 ± 0	0.02 ± 0.02
Peritoneal muscle	361.04 ± 148.75	56.83 ± 25.46	0 ± 0	0.04 ± 0.04
Neck muscle	392.29 ± 161.62	73.63 ± 32.98	0 ± 0	0.02 ± 0.02
***p*** **Value**	>0.05	>0.05	>0.05	>0.05
**B**				
**Additional Organ** **Tissues**	**Neu5Ac**		**KDN**	
	**Conjugated Form**	**Free Form**	**Conjugated Form**	**Free Form**
^2^ Brain	9527.89 ± 755.38	-	0 ± 0	-
Tongue	302.48 ± 130.37	67.63 ± 31.92	0 ± 0	0.71 ± 0.77
Rumen	881.83 ± 380.07	80.72 ± 38.10	0.01 ± 0.01	0.19 ± 0.20
Reticulum	919.66 ± 396.37	135.23 ± 63.83	0 ± 0	0.05 ± 0.05
Omasum	1456.8 ± 661.39	141.03 ± 70.52	0 ± 0	0.08 ± 0.09
Abomasum	2237.24 ± 964.25	151.71 ± 71.61	0 ± 0	12.83 ± 13.99
Small intestine	1774.01 ± 764.60	126.72 ± 59.81	0.01 ± 0	16.23 ± 17.69
Large intestine	1169.11 ± 481.67	161.90 ± 72.53	0 ± 0	8.51 ± 8.97
***p*** **Value**	*p* < 0.001	0.003	*p* < 0.001	*p* < 0.001

^1^ Values, expressed as µg/g protein, are means ± SEM. ^2^ Combination of ganglioside-conjugated and protein-conjugated Neu5Ac and KDN. Quantification of free Neu5Ac and KDN in brain tissue was not performed in this study.

## Data Availability

The data used to support the findings of this study can be made available by the corresponding author upon request.

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
