# Peer review of "Red Meat Derived Glycan, N-acetylneuraminic Acid (Neu5Ac) Is a Major Sialic Acid in Different Skeletal Muscles and Organs of Nine Animal Species—A Guideline for Human Consumers"

_foods, 2023, doi:10.3390/foods12020337_

Round 1

Reviewer 1 Report

The manuscript entitled „Red Meat Derived Human Glycan, N-acetylneuraminic acid (Neu5Ac) is a Major Sialic Acid in Different Skeletal Muscles and Organs of Nine Animal Species- A guideline for Human Consumers” presents interesting issue, but some problems should be corrected.

Authors in their study declared to assess “different skeletal muscles and organs of nine animal species- a guideline for human consumers of red meat”, however in my opinion the serious problem with the presented study results from those presented examples of “red meat” with “guidelines for human consumers”. Authors presented unusual choice of red meats that in their opinion requires guidelines or consumers, namely: cattle, sheep, goat, pig, deer, horse, kangaroo, dog, cat. While there are some commonly consumed types of animal/ meat (cattle, sheep, goat, pig), the other which are controversial but still are consumed in some populations (deer, horse, kangaroo), and the other which are cultural taboo (dog, cat) and can not be treated as “red meat” to be consumed and to require guidelines for human consumers.

Presenting the issue in such way is highly controversial and may be perceived as an ethical problem (as Authors in fact declare in their title that they present guidelines for human consumers of dogs and cats). Especially if slaughter, sale, purchase (including import), or consumption of dog meat is banned in some countries and legal in others, and cat meat consumption is eve more unusual.

Taking into account specific sentences from the manuscript, Authors should not present such perspective, as “red meat […] including female deer organs, and muscles from kangaroo and dog, should be the first choice for human consumption”.

Authors should get familiar with the currently indicated issues associated with sustainable consumption. Recommending people to consume above all “red meat” from kangaroo and dog is not in agreement with a current state of knowledge and such opinions should not be promoted.

Reviewer 2 Report

In this manuscript, Marefa et al. performed a quantitative analysis of Neu5AC and KDN in the skeletal muscle and organs across nine animal species. By integrating previously published Neu5Gc data, the authors made a comprehensive comparison of these sialic acids among different species. Overall, the manuscript is well written. There are still a few concerns that need to be addressed.   

Major:

The tissue samples used in this manuscript are at least 2 years away from the one used in Food Chem (2020) if I understand the processing correctly. To convince the readers of the quality of samples, could authors add a statement in section 3.3 that samples used for Neu5Ac, KDN, and Neu5Gc quantitation are from the same source? Here, the authors integrated the previous Neu5GC data with new quantitation data of Neu5Ac and KDN, which is established at the condition that there is no degradation of Neu5GC-related conjugates during sample storage. Could the authors confirm that the quantitation of Neu5Gc from the current batch is consistent with the one published in 2020?

Minor:

1.       In the abstract, only the conclusive sentence mentioned “Neu5Gc”, which makes readers difficult to understand why showing Neu5GC here. Could authors add more text related to Neu5GC in the abstract? Also, could the authors add a few sentences to establish the connection between meat safety and sialic acids?

2.       Line 93: A space should be between a value and a unit.

3.       Could authors add a few sentences regarding choosing different preparation methods for non-brain and brain samples?

4.       Figures 1 and 3 should be resized.

5.       Please add the replicate number in the legend of Figures 1 and 3.

6.       Line 148: Please correct the temperature unit.

7.       Please check the reference format.

Reviewer 3 Report

In this article by Jahana et al., the authors have determined the relative concentrations of two sialic acids in the different tissues of animals that serve as red meat food source for the consumers. They used biochemical methods to quantify the free and glycoconjugated Neu5Ac and Kdn amounts in these tissues. Based on their data, they also provided recommendations for the safe consumption of different red meats. Overall, the study is nicely done, and the conclusions supported by their data. There are a few comments regarding the scientific correctness of the narrative in the text that need to be addressed to avoid misinformation.

1.     Throughout the manuscript the authors have described Neu5Ac as the human (human-specific) sialic acid, which is scientifically incorrect. Neu5Ac is present in all vertebrates. In fact, humans are unique because they predominantly contain Neu5Ac in their glycans (and unlike other mammals cannot endogenously synthesis Neu5Gc, as also mentioned by the authors in the manuscript). Therefore, the authors should be cautious while using the term “human sialic acid, Neu5Ac”.

2.     Abstract – Line 38 - Neu5Gc is not carcinogenic, however it has been proposed to aggravate the cancer progression in humans. The authors have this correctly discussed in their discussion.

3.     Line 45 – Neu5Gc can be found in normal human tissues, however humans cannot synthesis the Neu5Gc endogenously. They incorporate Neu5Gc from the dietary sources. The cancerous tissues are hypersialylated, including containing more Neu5Gc than non-cancerous tissues.  

4.     Line 148 – the correct temperature should probably be 4℃, instead of 40 C.

5.     Section 2.6 – The standard curve for Neu5Gc is missing from the current data.

6.     Line 489 – This sentence is misleading. The author probably meant to say that the distribution and concentrations of Neu5Ac and Kdn containing sialoglycoconjugates in red meat…

7.     Line 494 – “…contains Neu5Ac, but not Kdn” – This is incorrect. There is the presence of free Kdn in high concentration, however it is the glycoconjugated Kdn which is absent unlike Neu5Ac glycans. Kdn is present primarily as the free sialic acid.

8.     Discussion, Line 523 - In this respect, it is to be noted that humans have recently been shown to contain naturally occurring antibodies against the Kdn glycans like Neu5Gc antibodies. So, the role of any possible Kdn incorporation on the human tissues also needs to be studied further.

9.     Overall, the authors should be cautious in describing the glycoconjugated (or bound) and free sialic acids because each of them has different physiological consequences as have been described in several publications.

Round 2

Reviewer 1 Report

The manuscript entitled „Red Meat Derived Human Glycan, N-acetylneuraminic acid (Neu5Ac) is a Major Sialic Acid in Different Skeletal Muscles and Organs of Nine Animal Species- A guideline for Human Consumers” presents interesting issue, but some problems should be corrected.

The problems which I indicated in my previous review are not solved. My major doubts are associated with the fact that Authors have studied dog and cat meat and ac a conclusion recommends to consume them. The response letter did not solve the problem as well. Authors indicated here that “in all states of Australia, with the exception of South Australia, it is not illegal to consume cat and dog meat. […] There are also no legal consequences if a pet owner kills and eats their dog or cat.”. In my opinion it still does not justify recommending to consume, as Authors do in their manuscript (“red meat […] including female deer organs, and muscles from kangaroo and dog, should be the first choice for human consumption”).

As I have indicated in my previous review:

Authors in their study declared to assess “different skeletal muscles and organs of nine animal species- a guideline for human consumers of red meat”, however in my opinion the serious problem with the presented study results from those presented examples of “red meat” with “guidelines for human consumers”. Authors presented unusual choice of red meats that in their opinion requires guidelines or consumers, namely: cattle, sheep, goat, pig, deer, horse, kangaroo, dog, cat. While there are some commonly consumed types of animal/ meat (cattle, sheep, goat, pig), the other which are controversial but still are consumed in some populations (deer, horse, kangaroo), and the other which are cultural taboo (dog, cat) and can not be treated as “red meat” to be consumed and to require guidelines for human consumers.

Presenting the issue in such way is highly controversial and may be perceived as an ethical problem (as Authors in fact declare in their title that they present guidelines for human consumers of dogs and cats). Especially if slaughter, sale, purchase (including import), or consumption of dog meat is banned in some countries and legal in others, and cat meat consumption is eve more unusual.

Taking into account specific sentences from the manuscript, Authors should not present such perspective, as “red meat […] including female deer organs, and muscles from kangaroo and dog, should be the first choice for human consumption”.

Authors should get familiar with the currently indicated issues associated with sustainable consumption. Recommending people to consume above all “red meat” from kangaroo and dog is not in agreement with a current state of knowledge and such opinions should not be promoted.

Author Response

Many thanks for your comments, point taken, we have revised our manuscript by adding one sentence to state that “dog and cat meat is not a conventional red meat consumed by humans globally” in the abstract and main text, please see the revised manuscript highlighted yellow.

Please note we do not state that kangaroo meat is not conventional red meat because the Australian government's new red meat initiative is to promote kangaroo meat consumption nationally and internationally. Kangaroo meat is produced in Australia from wild kangaroos and is exported to over 60 overseas markets (Government, Australian (4 November 2020) "Exporting kangaroo meat").  Kangaroo meat was legalized for human consumption in Australia in 1980. Many Australian supermarkets now stock various cuts of kangaroo (1-2) including fillets, steaks, minced meat, and 'Kanga Bangas' (kangaroo sausages). Many Australian restaurants serve kangaroo meat (3).

References:

  1. Dow, Steve (26 September 2007). "An industry that's under the gun". Sydney Morning Herald. Retrieved 19 August 2008.
  2. Benn, Matthew (4 September 2005). "Kangaroo meat exports jump even as drought culls supply". The Sun-Herald. Retrieved 21 August 2008.
  3. Rebecca Levingston (10 February 2010). "Kangatarianism – roo stew?". ABC Brisbane. Archived from the original on 31 August 2011. Retrieved 17 January 2012.
